# A Simple Baseline for Knowledge-Based Visual Question Answering

**Alexandros Xenos**[1*] **Themos Stafylakis**[2,3] **Ioannis Patras**[1] **Georgios Tzimiropoulos**[1]

[1]Queen Mary University of London     [2]Athens University of Economics and Business
[3]Omilia - Conversational Intelligence, Athens, Greece

{a.xenos,i.patras,g.tzimiropoulos}@qmul.ac.uk, tstafylakis@aueb.gr

## Abstract

This paper is on the problem of Knowledge-Based Visual Question Answering (KB-VQA). Recent works have emphasized the significance of incorporating both explicit (through external databases) and implicit (through LLMs) knowledge to answer questions requiring external knowledge effectively. A common limitation of such approaches is that they consist of relatively complicated pipelines and often heavily rely on accessing GPT-3 API. Our main contribution in this paper is to propose a much simpler and readily reproducible pipeline which, in a nutshell, is based on efficient in-context learning by prompting LLaMA (1 and 2) using question-informative captions as contextual information. Contrary to recent approaches, our method is training-free, does not require access to external databases or APIs, and yet achieves state-of-the-art accuracy on the OK-VQA and A-OK-VQA datasets. Finally, we perform several ablation studies to understand important aspects of our method. Our code is publicly available at *https://github.com/alexandrosXe/A-Simple-Baseline-For-Knowledge-Based-VQA*

## 1 Introduction

Knowledge-based VQA (KB-VQA) is a recently introduced VQA task (Wang et al., 2017, 2018; Marino et al., 2019; Shah et al., 2019) where the image alone is not sufficient to answer the given question, but effective utilization of external knowledge resources is additionally required. To solve such a task, a model would need not only strong visual perception but also reasoning capabilities while also being able to effectively incorporate world knowledge from external KBs (e.g. Wikipedia, etc) and LLMs. Systems capable of answering general and diverse questions about the visual world find a wide range of applications: from personal assistants to aids for the visually impaired and robotics [1].

Recently, several works on KB-VQA (Gui et al., 2022; Lin et al., 2022) have emphasized the significance of incorporating both explicit and implicit knowledge. However, such approaches usually require complicated pipelines. Firstly, a KB (e.g. wikidata) covering world knowledge needs to be maintained and used for knowledge retrieval which is time-consuming and very sensitive to noise. Secondly, powerful LLMs such as GPT-3 (Brown et al., 2020) or OPT-175B (Zhang et al., 2022) are leveraged due to the huge amount of implicit knowledge stored in their parameters and their powerful reasoning capabilities through few-shot in-context learning. However, the computational or even actual monetary cost (e.g. cost for API access) associated with accessing such models renders them unaffordable for many researchers. Thirdly, it is crucial to train a fusion mechanism that can effectively reason by combining the retrieved explicit and implicit knowledge.

**Main contributions:** We present a simple yet powerful pipeline for KB-VQA which by-passes the need for using most of the components of the above-mentioned systems. Specifically, the proposed system is simply based on few-shot prompting of LLaMA-13B (Touvron et al., 2023a,b). The *key component* of our method is the implementation of effective in-context learning using *question-informative captions as contextual information* which, as we show, results in large accuracy boosts.

The proposed system features several advantages: (1) it is entirely training-free, requiring only a few examples for in-context learning; (2) it is based on the open-source LLaMA-13B (Touvron et al., 2023a,b) (considerably smaller than the widely-used GPT-3); (3) it is straightforward to reproduce; and (4) achieves state-of-the-art (SOTA) accuracy on the widely-used OK-VQA (Marino et al., 2019) and A-OK-VQA datasets (Schwenk et al., 2022).

---

[*]Corresponding author.
[1]https://www.adelaide.edu.au/aiml/our-research/machine-learning/vqa-vision-and-language

## 2 Related Work on KB-VQA

**Methods Without LLMs:** Several methods have been proposed including KRISP (Marino et al., 2021) which uses a multi-modal pretrained BERT (Devlin et al., 2019), MAVEx (Wu et al., 2022) which proposes to validate promising answer candidates based on answer-specific knowledge retrieval, and DPR which uses pseudo-relevance labels integrated with answer generation for end-to-end training. Typically, these systems are not as competitive as the ones based on LLMs.

**Methods based on LLMs:** PICa (Yang et al., 2022) is the first method to adopt GPT-3 for solving the KB-VQA task in a few-shot manner by just providing a few in-context VQA examples. Gui et al. (2022) proposed to use both implicit (i.e. GPT-3) and explicit (i.e. KBs) knowledge based on CLIP retrieval (Radford et al., 2021) which are combined by a novel fusion module called KAT (based on T5 or Bart). Lin et al. (2022) proposed to integrate local visual features and positional information (bounding box coordinates), retrieved external and implicit knowledge (using a GPT-3) into a transformer-based question-answering model. Hu et al. (2023) proposed PromptCap, a novel task-aware captioning model that uses a natural language prompt to control the generation of the visual content that can be used in conjunction with GPT-3 in-context learning. Img2Prompt Guo et al. (2023) is a zero-shot VQA method that generates image-relevant exemplar prompts for the LLM. Their key insight is that synthetic question-answer pairs can be generated using image captioning and question-generation techniques as in-context exemplars from the provided image. Prophet Shao et al. (2023) proposes to prompt GPT-3 with answer heuristics (answer candidates and answer-aware examples) that are encoded into the prompts to enable GPT-3 to better comprehend the task, thus enhancing its capacity.

## 3 Methodology

While explicit knowledge retrieval focuses on semantic matching between an image and knowledge entries, it lacks implicit commonsense knowledge (e.g. Lemons are sour) which can be found in LLMs (Gui et al., 2022). LLMs are critical in extracting implicit knowledge due to the vast amount of implicit information embedded in their parameters, and their powerful reasoning capacity through few-shot in-context learning. Different from pre-

vious work (Yang et al., 2022; Gui et al., 2022; Lin et al., 2022) we leverage the open-source LLM LLaMA-13B (Touvron et al., 2023a,b) instead of GPT-3 as an implicit language knowledge base and treat VQA as an open-ended text generation task.

Our method builds upon the pipeline of PICa, which is the pioneering work that utilizes GPT-3 for few-shot in-context learning in order to address the KB-VQA task. GPT-3 is a decoder-only autoregressive LLM of 175B parameters, trained on a diverse range of data sources, including Common Crawl, webtexts, books, and Wikipedia (Brown et al., 2020). During inference, in-context few-shot learning involves formulating a novel downstream task as a text sequence generation task using the frozen GPT-3 model. When provided with a testing input $x$, the target $y$ is predicted based on a formatted prompt $p(h, C, E, c, x)$. In this prompt, $h$ represents a prompt head or instruction that describes the task, while $E = \{e_1, e_2, ..., e_n\}$ represents a set of $n$ in-context examples (shots), where $e_i = (x_i, y_i)$ represents an input-target pair of the task, where $x_i$ and $y_i$ are the input and target, respectively. These pairs are constructed manually or sampled from the training set. $C = \{c_1, c_2, ..., c_n\}$ represents a set of generic image captions describing each $x_i$ since images cannot be inputted to GPT-3. The caption for the test input is labeled as $c$. The target $y$ is denoted as a text sequence consisting of $L$ tokens, expressed as $y = (y^1, y^2, ..., y^L)$. At each decoding step $t$, the following conditions apply:

$$\hat{y}^t = \underset{y^t}{\arg\max}\, p_{LLM}(y^t | p, \hat{y}^{<t}) \qquad (1)$$

In order to utilize any LLM for the knowledge-based VQA task, the crucial step is to design suitable prompts. When given a question $q_i$ and an image $v_i$ as inputs, the VQA task's objective is to predict the corresponding answer $a_i$. However, since LLMs do not inherently comprehend images, it becomes necessary to convert the image into a caption $c_i$ using a pre-existing captioning model. While SOTA pretrained captioning models have demonstrated impressive performance, they are primarily optimized to generate generic image captions. Unfortunately, these captions often fail to capture all the specific details required to accurately answer a given question about the image. In this work, instead of generic captions, we generate question-guided informative image captions using the Plug-and-Play VQA (PNPVQA) framework (Tiong et al., 2022) which identifies the most re-

lated image patches to the question with a saliency map-based interpretability technique and generates captions from these patches only.

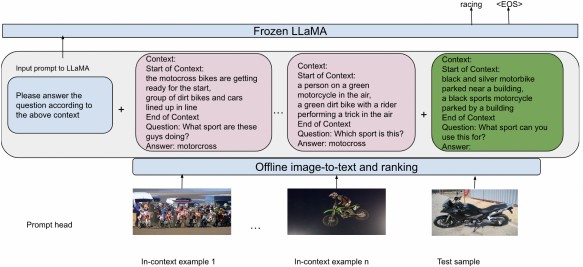

Figure 1: Inference-time of our method for n-shot VQA. The input prompt to LLaMA consists of a prompt head $h$ (blue box), $n$ in-context examples ($\{c_i, x_i, y_i\}_{i=1}^{n}$) (red boxes), and the VQA input $\{c, x\}$ (green box). The answer $y$ is produced in an open-ended text generation manner. In this example we use two question-informative captions per example (separated by commas).

For each image-question pair, we first generate 50 question-guided informative image captions from the image $v_i$ using PNPVQA. We then employ BLIP's (Li et al., 2022) text encoder to encode all the image captions and BLIP's image encoder to encode the image $v_i$. We rank the image captions per image $v_i$ according to their cosine similarity with the image $v_i$ and keep the top-$m$ most similar captions $c_i$ per example. After extracting the top-$m$ most similar captions per image $v_i$ we construct a carefully designed text prompt consisting of a general instruction sentence, the captions $C$, the question, the test input's captions $c$, and a set of context-question-answer triplets (shots) taken from the training dataset that are semantically most similar to the current image-question pair (see Fig. 1). Then this text prompt is passed to a frozen LLaMA-13B model and in-context few-shot learning is performed in order to obtain its output as a promising answer candidate to the current image-question pair.

### 3.1 Selecting Informing Examples For Few-Shot In-Context Learning

As Yang et al. (2022) notes, feeding more in-context examples to GPT-3 yields better few-shot performance. However, the maximum input length of the model constrains the maximum number of examples $n$ in the prompt. To better use these available examples we: (i) improve the example quality by careful in-context example selection (Liu et al., 2022; Gui et al., 2022; Shao et al., 2023), and (ii)

| Method | Knowledge Resources | Acc (%) |
|---|---|---|
| KRISP | Wikipedia+ConceptNet | 38.35 |
| MAVEx | Wikipedia+ConceptNet+Google Images | 39.4 |
| Unified-IO (2.8B) | Multimodal Pretraining | 54 |
| Flamingo (80B) | Multimodal Pretraining | 57.8 |
| PICa-Full | Frozen GPT-3 (175B) | 48.0 |
| KAT_base (single) | Wikidata+Frozen GPT-3 (175B) | 50.58 |
| KAT_large (single) | Wikidata+Frozen GPT-3 (175B) | 53.09 |
| KAT_large (ensemble) | Wikidata+Frozen GPT-3 (175B) | 54.41 |
| REVIVE_large (single) | Wikidata+Frozen GPT-3 (175B) | 56.6 |
| REVIVE_large (ensemble) | Wikidata+Frozen GPT-3 (175B) | 58.0 |
| Prophet | Frozen GPT-3 (175B) | 61.1 |
| Ours | Frozen LLaMA (13B) | 58.69 |
| Ours + MCAN | Frozen LLaMA (13B) | 60.02 |
| Ours | Frozen LLaMA 2 (13B) | 59.07 |
| **Ours + MCAN** | **Frozen LLaMA 2 (13B)** | **61.2** |

Table 1: Comparison with other methods on the OK-VQA dataset: Our method with 9 question-informative captions achieves state-of-the-art performance.

use more examples via multi-query ensemble.

**In-context Example Selection** tries to search for the best examples for each inference-time input $x$ among all available examples (Yang et al., 2022). We consider in-context examples that have similar question features as $x$. More specifically, given an inference-time question, we use BLIP's text encoder to obtain its textual feature and compute its cosine similarity with the questions in all available in-context examples. We then average the question text similarity with the image visual similarity to guide the example selection similarly to Yang et al. (2022). We select the top-$n$ questions with the highest similarity and use the corresponding examples as the in-context examples.

**Multi-query ensemble:** Given an inference-time example $x$, we use $k \times n$ in-context examples to generate $k$ prompts. This way, we prompt LLaMA-13B for $k$ times and obtain $k$ answer predictions instead of 1 similar to Yang et al. (2022), where $k$ is the number of queries to ensemble. Finally, among the $k$ answer predictions, we select the one with the most occurrences (majority vote).

## 4 Experimental Results

**Comparative results on OK-VQA:** Table 1 summarizes the results of various methods on OK-VQA including our best method (last row) which uses 9 question-informative captions and 5 query ensembles. When using LLaMA our approach outperforms all methods and achieves comparable results with Prophet especially when using the same shot selection strategy based on MCAN (Yu et al., 2019). Moreover, it performs better than Unified-IO and the 80B Flamingo which have been pre-trained

| Method | DA | | MC | |
|---|---|---|---|---|
| | Val | Test | Val | Test |
| ClipCap | 30.9 | 25.9 | 56.9 | 51.4 |
| ViLBERT | 30.6 | 25.9 | 49.1 | 41.5 |
| LXMERT | 30.7 | 25.9 | 51.4 | 41.6 |
| KRISP | 33.7 | 27.1 | 51.9 | 42.2 |
| GPV-2 | 48.6 | 40.7 | 60.3 | 53.7 |
| Unified-IO | - | 45.2 | - | - |
| Prophet | 58.2 | 55.7 | 59.3 | 57.3 |
| Ours (LLaMA) | 54.4 | 53.8 | - | - |
| Ours + MCAN (LLaMA) | 57.4 | 55.0 | - | - |
| Ours (LLaMA 2) | 57.1 | 55.4 | - | - |
| **Ours + MCAN (LLaMA 2)** | **58.6** | **57.5** | - | - |

Table 2: Comparison with other methods on the A-OK-VQA dataset: Our method with 9 question-informative captions achieves state-of-the-art performance at the direct answer (DA) setting. Note that our method does not support multiple-choice (MC).

with multimodal objectives. When compared to methods that rely on GPT-3 for implicit knowledge extraction, our approach outperforms PICa-Full which only uses generic image captions by 12.02% while outperforming the SOTA supervised methods KAT and REVIVE by 5.61% and 2.02% respectively. Finally, when using LLaMA 2 and MCAN-based shot selection strategy, our method achieves state-of-the-art accuracy of 61.2%.

**Comparative results on A-OK-VQA:** Table 2 summarizes the results of various methods on A-OK-VQA including our best method (last row) which uses 9 question-informative captions and 5 query ensembles. We compare our method to the strong baselines in (Schwenk et al., 2022) and the current state-of-the-art method Prophet (Shao et al., 2023). When employing LLaMA, our approach surpasses all other methods on the DA setting and achieves comparable results to Prophet, particularly when employing the same shot selection strategy based on MCAN. Finally, with LLaMA 2 and MCAN our method attains state-of-the-art performance on both the validation and test sets, achieving 58.6% and 57.5% accuracy respectively, demonstrating the effectiveness and robust generalization of our proposed method.

## 5 Ablation Studies

We conduct several ablations on OK-VQA to better understand the key components of our method.

**Effect of question-informative captions:** Table 3

| Captions | $n$ | $k$ | Acc (%) |
|---|---|---|---|
| Generic | 14 | 5 | 43.35 |
| Question-informative | 14 | 5 | 57.56 |

Table 3: Generic vs. question-informative captions.

shows the performance of our method when using generic captions vs question-informative captions for in-context learning which is the key component of our system. Following Yang et al. (2022); Shao et al. (2023) we leverage the OSCAR+ (Zhang et al., 2021) as the captioning model. The results suggest using question-informative captions results in huge accuracy boosts (43.35% vs 57.56%).

| Shot Selection Strategy | Captions | $m$ | $n$ | $k$ | Acc (%) |
|---|---|---|---|---|---|
| Random | Question-informative | 1 | 14 | 5 | 53.19 |
| Avg. Question and Image Sim. | Question-informative | 1 | 14 | 5 | 56.50 |
| MCAN latent space | Question-informative | 1 | 14 | 5 | 57.56 |

Table 4: Accuracy when using different shot selection strategies. Avg. question and image sim. strategy retrieves shots based on the average cosine similarity between the test sample's question and image, and the training examples' question and image. MCAN latent space strategy retrieves shots that are closer to the test sample in the trained MCAN's latent space.

**Effect of shot selection strategy:** Table 4 shows that selecting random shots during in-context learning hurts the accuracy, confirming the findings of Yang et al. (2022). Retrieving shots based on the similarity between the test sample and the training examples yields a significant accuracy boost. Prophet's shot selection strategy based on MCAN also seems to be effective but we note that it is based on pre-training a vanilla VQA model on a different dataset (VQA-v2).

**Effect of number of question-informative captions:** Fig. 2 (a) shows the accuracy when we increase the number of captions per sample in the prompt during in-context learning. Here, we are using $k = 5$, and $n = 10$ when using 1-10 captions, and $n = 5$ when using more than 10 captions due to max. sequence length constraints. More captions provide more information for each example helping the model to make a more accurate prediction based on context. As shown in the figure, the validation accuracy keeps increasing up to 60.02%. When using more than 10 captions, the accuracy decreases but this also can be attributed to the fact that we are also decreasing

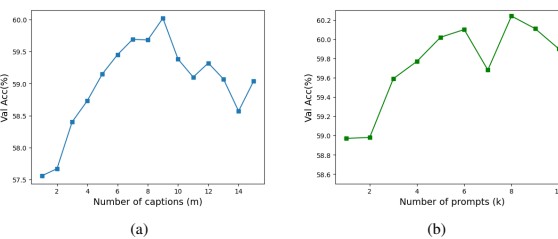

(a)                    (b)

Figure 2: (a) Accuracy vs number of question informative captions used per shot during few shot in-context learning. (b) Accuracy vs number of prompts $k$ used during in-context learning.

the number of shots to 5.

**Effect of multi-query ensemble:** Fig. 2 (b) shows the accuracy as the number of prompts, $k$, increases. As anticipated, employing multiple prompts of LLaMA instead of just one yields improved accuracy. However, beyond $k = 6$, the accuracy begins to fluctuate. It is important to note that this fluctuation could be attributed to the retrieval of noisy (irrelevant to the question) context examples as the value of $k$ increases.

**Effect of explicit knowledge:** We also tried to use KAT's (Gui et al., 2022) KB and trained a T5 (Raffel et al., 2020) in order to integrate explicit knowledge into our model. For each image, we used BLIP to extract explicit knowledge via image-to-text retrieval. We used $40$ retrieved passages and LLaMA predictions as explicit and implicit knowledge, respectively. We achieved an accuracy of 58.70% which shows that our model does not benefit from such an approach.

**Effect of size of LLM:** We also used a LLaMA-7B model using 9 question-informative captions, $n = 10$ and $k = 5$. Reducing the size of the LLM leads to decreased accuracy but the drop is not large, still obtaining 57.99% accuracy.

## 6 Conclusions

We proposed a simple yet effective baseline for KB-VQA. Our training-free method is based on in-context few-shot learning of the open-source LLaMA using question-informative captions. We show that this is sufficient to achieve SOTA results on the widely used OK-VQA and A-OK-VQA datasets.

## Limitations

It is important to acknowledge that we have not explored the utilization of any other medium-sized

LLMs apart from LLaMA, which presents a limitation of our study. Lastly, due to limitations in resources, we were unable to conduct experiments with larger sizes beyond 13B. However, it would indeed be intriguing to observe the performance when employing LLaMA models of sizes such as 30B or 65B.

## Ethics Statement

The authors of this paper recognize the importance of responsible AI in both research and development endeavors. We are committed to ensuring that the model we develop is not only accurate but also fair and unbiased. We understand the potentially significant impact of VQA technology on society and, therefore, pledge to maintain transparency by sharing our findings and progress with relevant researchers and stakeholders.

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

## A  Example Appendix

### A.1  Implementation Details

We used the Huggingface Transformers library[2] in order to run LLaMA models. We used beam search with beam size = 2 during generation and max new tokens = 5 while using the default values for all the other parameters in the generate method. We run our model on a 40-GB VRAM A-100 GPU card.

---

[2]https://huggingface.co/transformers/