# OpenReview forum: "A Simple Baseline for Knowledge-Based Visual Question Answering"
_EMNLP/2023/Conference — EMNLP 2023 Main_

### Official Review · Reviewer_z51R · 2023-07-24

**Typos Grammar Style And Presentation Improvements:** Overall the paper reads well
**Soundness:** 4

**Excitement:**

4: Strong: This paper deepens the understanding of some phenomenon or lowers the barriers to an existing research direction.

**Missing References:**

It is odd that the KAT model is not introduced in the related work but that presented in the table of results.

The paper uses MCAN but does not explain or provide a reference for this technique

The idea of using captions and specifically question-focused captions for VQA were first introduced by papers in ACL 2019 by Kim and Bansal, and by Wu and Mooney (question specific), and it would have been nice to reference this earlier related work.

**Paper Topic And Main Contributions:**

The paper presents a simpler model that relies on a smaller opensource LLM that achieves close to SOTA performance on the problem of Knowledge-Based Visual Question Answering (KB-VQA)  using few-shot learning that exploits question-focused captioning to provide image info to the LLM

**Questions For The Authors:**

What is the real algorithmic novelty of this work since it is a straightforward combination of existing methods?

**Reasons To Accept:**

The paper quite clearly presents a fairly simple new baseline system for the important problem of KB-VQA that achieves near SOTA performance on the OK-VQA standard benchmark for this task, almost tying the recent PROPHET system that uses GPT-3.  It combines a smaller open LLM (LLaMA) and a question focused captioner from the PNPVQA system.  The model is simpler and more open than current SOTA models for OK-VQA that use a closed, larger GPT-3 model.  Therefore, it provides a new more accessible and efficient model for KB-VQA.  The paper provides a series of ablation studies exploring the contribution of various aspects of the model such as the question-focusing of the captioner, few shot sample selection, the size of the few-shot training set, model size, and number of provided captions. Overall, a solid short paper on an important topic. The fact than a modest-sized model without explicit knowledge retrieval can do near-SOTA on this problem is somewhat surprising and interesting.

The rebuttal results with LLaMA that achieve now SOTA results is a nice addition, I raised my excitement score to a 4, but if it was an option, I would have given a 3.5, since my excitement has only increased a relatively small amount based on this.

**Reasons To Reject:**

The paper does not really introduce any new methods or algorithms, just combining existing methods like LLaMA and PNPVQA. So there is very little if any algorithmic novelty.  The results are close but do not actually surpass the SOTA results from Prophet.  So the only real advantages are simplicity, openness and efficiency.

Results on larger models and now LLaMA 2 would be nice, to see if this approach could actually achieve SOTA with a larger more modern model.

The paper is solid but not very novel or exciting

**Reproducibility:**

4: Could mostly reproduce the results, but there may be some variation because of sample variance or minor variations in their interpretation of the protocol or method.

**Reviewer Confidence:**

4: Quite sure. I tried to check the important points carefully. It's unlikely, though conceivable, that I missed something that should affect my ratings.

---

> ### Author Rebuttal · Authors · 2023-08-29
>
> Q3.1: “Results on larger models and now LLaMA 2 would be nice, to see if this approach could actually achieve SOTA with a larger more modern model.”
>
> A3.1: When we were submitting the paper LLaMa 2 was not out yet. We were able to run LLaMA 2 now and got impressive results even surpassing the state-of-the-art model PROPHET. More specifically, using LLaMa 2 + MCAN we got 61.2% validation accuracy, while using LLaMa 2 without MCAN resulted in 59.7% validation accuracy. We will add these results to the updated version of our paper. Thank you.
>
> Q3.2: “What is the real algorithmic novelty of this work since it is a straightforward combination of existing methods?”
>
> A3:2 Our method is simple but not simplistic and it is proposed for the first time to address this particular problem. We are also the first to show that such a simple approach can be used to produce competitive results (or even SOTA results with Llama2; see A3.1) to recent SOTA approaches based on much more complicated pipelines. The method is based on in-context learning but the context is not just a generic image caption, it contains rich question-informative captions (after ranking them using text-to-image similarity) that describe the image with respect to the question. Furthermore, adding more captions per shot results in better describing the image for the LLM to perform the QA task, while encouraging diversity of captions and coverage of visual content.
>
> Q3.3 “It is odd that the KAT model is not introduced in the related work but that presented in the table of results.
> The paper uses MCAN but does not explain or provide a reference for this technique
> The idea of using captions and specifically question-focused captions for VQA were first introduced by papers in ACL 2019 by Kim and Bansal, and by Wu and Mooney (question specific), and it would have been nice to reference this earlier related work.”
>
> A3.3: We would like to thank you for pointing this out, we will of course address this in the updated version of our paper. Thank you again.

---

### Official Review · Reviewer_3YRF · 2023-08-04

**Soundness:** 4

**Excitement:**

3: Ambivalent: It has merits (e.g., it reports state-of-the-art results, the idea is nice), but there are key weaknesses (e.g., it describes incremental work), and it can significantly benefit from another round of revision. However, I won't object to accepting it if my co-reviewers champion it.

**Paper Topic And Main Contributions:**

This paper focuses KB-VQA. Despite the current emphasis on using both explicit and implicit knowledge to answer questions needing external data, there are often complex pipelines and reliance on LLM API. This paper introduced a simpler, easily replicable process centered on efficient in-context learning and open-source LLaMA, negating the need for training or access to external databases or APIs. Despite this, the method maintains a competitive accuracy rate on the OK-VQA dataset. The paper concludes with several ablation studies for a deeper understanding of the method's key components.

**Questions For The Authors:**

A. As the research of LLM advances, LLaMA series model had reached similar performance with older GPT-3, what would the method perform with the proposed prompting mechanisms applied on GPT-3 (as used in compared methods) for fair comparison?

**Reasons To Accept:**

The paper introduced a reproducible KB-VQA approach using a LLM (i.e., LLaMA) as knowledge source. They designed interesting caption generation and in-context demonstration selection mechanisms to improve the few-shot performance from LLM, especially on generating question-related captions. Despite the manuscript's concise length, the conducted experiments are thorough, effectively substantiating the efficacy of the devised mechanisms. Remarkably, the performance proves competitive even when compared to methodologies leveraging larger LLMs, such as GPT-3.

**Reasons To Reject:**

While the few-shot prompting mechanisms demonstrate notable efficacy, it is worth noting their resemblance to the methodology employed in Prophet, which also utilizes a similarity-based sample selection mechanism.
It may be beneficial for the author to confirm the effectiveness of their approach using GPT-3, as is the case in other comparative methodologies. This could provide a more accurate assessment of the benefits gained from the proposed prompting mechanism. Furthermore, the reviewer anticipates a comparative analysis across other KBVQA datasets, such as A-OKVQA, to broaden the evaluation of the method's applicability. The reviewer also expects the analysis of the inability of retrieved external knowledge to improve the model performance.
The rebuttal addressed some concerns above (A-OKVQA result and GPT-3 based result). However, if LLaMA is already better than GPT-3 (on KBVQA as demonstrated in performance comparison in rebuttal, and on pure NLP tasks as demonstrated in LLaMA paper), the performance improvement might simply come from a better (although less parameters) LLM used.

**Reproducibility:**

4: Could mostly reproduce the results, but there may be some variation because of sample variance or minor variations in their interpretation of the protocol or method.

**Reviewer Confidence:**

3: Pretty sure, but there's a chance I missed something. Although I have a good feel for this area in general, I did not carefully check the paper's details, e.g., the math, experimental design, or novelty.

---

> ### Author Rebuttal · Authors · 2023-08-29
>
> Q2.1: “Furthermore, the reviewer anticipates a comparative analysis across other KBVQA datasets, such as A-OKVQA, to broaden the evaluation of the method's applicability.”
>
> A2.1: We ran our model (without MCAN) to the newer  KB-VQA dataset A-OK-VQA and got a validation accuracy of 54.4% and a test accuracy of 53.8%. Our model again shows comparable performance to PROPHET which has a validation accuracy of 58.2% and a test accuracy of 55.7%. Thank you for this we will add it to the paper.
>
> Q2.2: “The reviewer also expects the analysis of the inability of retrieved external knowledge to improve the model performance.”
>
> A2.2: From the analysis we performed we found that adding explicit knowledge to the pipeline (through KAT) affects only a small sample of the predictions (486 out of the 5046 (9.63%) total predictions). In this small sample, the LLaMA model (implicit only) is most of the time correct while KAT is wrong (156 out of the 486 (32.1%) samples). Only in  82 out of 486 (16.87%) KAT was correct while LLaMA was wrong. Finally, in 98 out of 486 (20.16%) predictions, both models were correct with different predictions and in the last 150 out of the 486 total different predictions (30.86%), both models were wrong.
>
> Q2.3: “ As the research of LLM advances, LLaMA series model had reached similar performance with older GPT-3, what would the method perform with the proposed prompting mechanisms applied on GPT-3 (as used in compared methods) for fair comparison?”
>
> A2.3: Due to the high cost of openAI’s API we were able to run experiments using Instruct-GPT-3 (text-davinci-003) using only k = 1(number of prompts in the multi-query ensemble) and using 9 question-informative captions. The result we got was 55.34% validation accuracy which is smaller than the respective of our LLaMA-13B model in paper 58.97%. Importantly, our work focuses on open-sourced small-medium size LLMs.

---

### Official Review · Reviewer_P6LA · 2023-08-05

**Soundness:** 3

**Excitement:**

3: Ambivalent: It has merits (e.g., it reports state-of-the-art results, the idea is nice), but there are key weaknesses (e.g., it describes incremental work), and it can significantly benefit from another round of revision. However, I won't object to accepting it if my co-reviewers champion it.

**Paper Topic And Main Contributions:**

- The paper deal with the problem of knowledge-based visual question answering (KB-VQA).
- The authors propose fully white-box framework for KB-VQA based on LLaMA-13B model.
- The proposed framework achieves comparable performance with GPT-3 API-based VQA baselines and does not require any addional training steps.

**Questions For The Authors:**

- If possible, can you provide peformance after some fine-tuning steps? This can prove wider applicability of the proposed framework.
- [1] propose a new ensemble scheme for QA with few-shot examples. Can you provide performance of your framework equipped with idea of [1] in multi-query ensemble?
- If I understood correctly, we may replace LLaMA-13B model in the overall process of this paper into GPT-4, which is the state-of-the-art large language model. Since, this work utilizes LLM without any finetuning, I expect that GPT-4 can show upper-bound performance which is higher than LLaMA-13B. Even though GPT-4 is black-box model, I want to see the upper bound performance of the framwork.

[1] Replug: Retrieval-augmented black-box language models, Shi et al

**Reasons To Accept:**

- If I understood correctly, this work is the first LLM-based reproducible open-source VQA framework. Since VQA is one of the most practical problem in NLP era, this framework can be employed as basis for the future research on VQA.
- The empirical results show that the proposed VQA framework achieves SOTA peformance which is comparable with existing black-box framework based on GPT-3 for OK-VQA dataset.
- The proposed method is simple and intuitive.
- Nice ablation results (Section 5).

**Reasons To Reject:**

- I think novelty of the proposed method is not so significant. (However, it is not important for this kind of research.)
- Please refer to questions.

**Reproducibility:**

4: Could mostly reproduce the results, but there may be some variation because of sample variance or minor variations in their interpretation of the protocol or method.

**Reviewer Confidence:**

2: Willing to defend my evaluation, but it is fairly likely that I missed some details, didn't understand some central points, or can't be sure about the novelty of the work.

---

> ### Author Rebuttal · Authors · 2023-08-29
>
> Q1.1: “If possible, can you provide performance after some fine-tuning steps? This can prove wider applicability of the proposed framework”
>
> A1.1: We have tried to finetune LLaMA using parameter-efficient techniques but we still got an OOM (out-of-memory) error because the prompts are very lengthy (1856 subtokens per average). Thank you we will discuss this in the paper.
>
> Q1.2 “Replug: Retrieval-augmented black-box language models, Shi et al,  propose a new ensemble scheme for QA with few-shot examples. Can you provide performance of your framework equipped with idea of [1] in multi-query ensemble?”
>
> A1.2: The authors of the mentioned paper haven’t made the code available yet. We didn’t have the time to reproduce the code but we will try to address this for the final version of our paper!
>
> Q1.3: “If I understood correctly, we may replace LLaMA-13B model in the overall process of this paper into GPT-4, which is the state-of-the-art large language model.”
>
> A1.3: Yes this is true, we can replace LLaMA-13B model with any Large Language Model in the overall process of the paper.
>
> Q1.4: “Since, this work utilizes LLM without any finetuning, I expect that GPT-4 can show upper-bound performance which is higher than LLaMA-13B. Even though GPT-4 is black-box model, I want to see the upper bound performance of the framwork.”
>
> A1.4: Due to the high cost of openAI’s API we were able to run experiments using Instruct-GPT-3 (text-davinci-003 model) using only k = 1(number of prompts in the multi-query ensemble) and using 9 question-informative captions. The result we got was 55.34% validation accuracy which is smaller than the respective of our LLaMA-13B model in paper 58.97%. Importantly, our work focuses on open-sourced small-medium size LLMs.
> Thank you for this we will add it to the paper.
>
> Q1.5: “ Would be hard pressed to reproduce the results. The contribution depends on data that are simply not available outside the author's institution or consortium; not enough details are provided.”
>
> A1.5: We kindly disagree. The results are 100% reproducible and we will open-source the code for this framework. The data and all the models used are 100% open-sourced.

---

### Meta-Review · Area_Chair_bBGv · 2023-09-17

**Recommendation:** 4

**Metareview:**

The authors present a fairly simple new baseline system for the important problem of KnowledgeBased-VisualQA that achieves near SOTA performance on a popular benchmark.

Reasons to accept:
- The proposed method is simple and intuitive.
- The conducted experiments are thorough, effectively substantiating the efficacy of the devised mechanisms.
- The analysis and the ablation are robust.

---

### Decision · Program_Chairs · 2023-10-07

**Decision:**

Accept-Main

**Comment:**

The authors present a fairly simple new baseline system for the important problem of KnowledgeBased-VisualQA that achieves near SOTA performance on a popular benchmark.

Reasons to accept:
- The proposed method is simple and intuitive.
- The conducted experiments are thorough, effectively substantiating the efficacy of the devised mechanisms.
- The analysis and the ablation are robust.